# IMPROVED ROBUSTNESS AND HYPERPARAMETER SELECTION IN THE DENSE ASSOCIATIVE MEMORY

## ABSTRACT

The Dense Associative Memory generalizes the Hopfield network by allowing for sharper interaction functions. This increases the capacity of the network as an autoassociative memory as nearby learned attractors will not interfere with one another. However, the implementation of the network relies on applying large exponents to the dot product of memory vectors and probe vectors. If the dimension of the data is large the calculation can be very large and result in imprecisions and overflow when using floating point numbers in a practical implementation. We describe the computational issues in detail, modify the original network description to mitigate the problem, and show the modification will not alter the networks' dynamics during update or training. We also show our modification greatly improves hyperparameter selection for the Dense Associative Memory, removing dependence on the interaction vertex and resulting in an optimal region of hyperparameters that does not significantly change with the interaction vertex as it does in the original network. Our modifications also allow us to train a Dense Associative Memory with larger interaction vertices than have been used in any previous literature.

## 1 INTRODUCTION

Autoassociative memories are a class of neural networks that learn to remember states, typically also allowing nearby states to iterate towards similar learned states. These networks act as memories for the learned states, reconstructing lost information and correcting errors in probe states. The Hopfield network (Hopfield, 1982; 1984) is perhaps the most studied model in the class. However, as with all autoassociative memories, the Hopfield network suffers from capacity issues – the number of states that can be stored in a network without error is limited. In the Hopfield network with Hebbian learning, this has been shown to be roughly $0.14N$ for a network of dimension $N$ (McEliece et al., 1987; Hertz, 1991). The Dense Associative Memory, also known as the modern Hopfield network, generalizes the classical Hopfield network by introducing an interaction function parameterized by an interaction vertex (Krotov & Hopfield, 2016; 2018). This function controls the range of the influence for learned states, allowing control of the sizes of the attractors and increasing the network capacity. Krotov & Hopfield (2016) also introduce several other generalizations which are parameterized by additional hyperparameters relating to learning, including the initial learning rate, learning rate decay, momentum, learning temperature and the exponent on the error term. Additional hyperparameters were introduced such as the form of the interaction function, the number of memory vectors, and more. In effect, the Dense Associative Memory is a potentially more powerful autoassociative memory, but at the cost of increased complexity and reliance on hyperparameter tuning.

We focus on the implementation details of the Dense Associative Memory. In particular, we show the exact form given by Krotov and Hopfield suffers from issues relating to computation and numerical stability. This form calculates the dot product between two vectors of length $N$ then immediately applies a potentially large exponentiation based on the interaction function. This can cause inaccuracies in the floating point numbers

used for computation, or even completely overflow them. In Section 4 we show a modification to the original form – a normalization and shifting of scaling factors – that prevents the computational problems, and prove that the modifications do not change the network behavior for a specific class of interaction functions: homogenous functions. Fortunately, the typical interaction functions – the polynomial interaction function (Equation 7) and rectified polynomial interaction function (Equation 8) – are in this class. We show our modifications do not alter the properties of the autoassociative memory, such as the capacity, but do appear to have desirable effects on the network over the course of training. In Section 5 we provide experimental results that show our modified network has a stable region of optimal hyperparameters across a wide range of interaction vertices. This is in comparison to the original network which had the optimal hyperparameters shift dramatically as the interaction vertex changed even for the same dataset. We also show that the optimal region of hyperparameters is no longer heavily dependent on the size of the data vectors or the interaction vertex, meaning applying the Dense Associative Memory to a new task will not require massively retuning the hyperparameter selections. A comprehensive list of our results, shown in Appendix A and B, demonstrate successful training of a Dense Associative Memory with interaction vertices up to $n = 100$, an arbitrary stopping point with indications of higher interaction vertices being possible and stable. Current literature has not discussed using an interaction vertex this large.

Our modified update rule

$$
\xi_i^{(t+1)} = \text{sign}\left[\sum_\mu \left( F_n(\alpha(\zeta_i^\mu + \sum_{j\neq i} \zeta_j^\mu \xi_j^{(t)})) - F_n(\alpha(-\zeta_i^\mu + \sum_{j\neq i} \zeta_j^\mu \xi_j^{(t)})) \right) \right], \tag{1}
$$

and learning rule / loss function

$$
\mathcal{L} = \sum_a \sum_i (\xi_{a,i} - C_{a,i})^{2m}
$$

$$
C_{a,i} = \tanh\left[\sum_\mu \left( F_n(\beta(\zeta_i^\mu + \sum_{j\neq i} \zeta_j^\mu \xi_j^{(t)})) - F_n(\beta(-\zeta_i^\mu + \sum_{j\neq i} \zeta_j^\mu \xi_j^{(t)})) \right) \right], \tag{2}
$$

are subtly different to the original specifications by Krotov & Hopfield (2016), in that the scaling factors $\alpha, \beta$ are within the interaction function evaluations $F_n(\cdot)$. We suggest values of $\alpha = \frac{1}{N}$, $\beta = \frac{1}{NT}$, for network dimension $N$ and temperature $T$.

## 2 LITERATURE REVIEW

Our proposed method of shifting the scaling factors within the interaction function does not appear to have been suggested previously, and other implementations of the Dense Associative Memory do not seem to have included it. However, many implementations of the Dense Associative Memory use the feed-forward equivalence set forth by Krotov & Hopfield (2016). This equivalence allows the Dense Associative Memory to be expressed with some approximations as a feed-forward densely connected neural network with a single hidden layer. This architecture is much easier to implement using traditional deep learning software libraries. The feed-forward equivalent model implicitly implements our proposed changes by selecting values of the scaling factor that negate terms arising from a Taylor expansion. This may help explain why the feed-forward version of the model is more stable and popular than the auto-associative version.

Normalization is a typical operation in neural networks. In autoassociative memories specifically, we may apply a normalization term to provide a constant power throughout network calculations, which ensures calculations are proportional only to the magnitudes of the learned weights rather than the magnitude of the probe vector. Even more specifically, in the Hopfield network this is typically achieved by using binary

valued vectors. It has been shown networks using these vectors have the same behavior as networks using graded (continuous value) neurons (Hopfield, 1984). Normalization may also be applied in the learning rule, such as in the Hebbian learning rule (Hebb, 1949). Normalization in learning may be used to simply scale the weights into something more interpretable, as in the Hebbian, or to achieve a different behavior during training. For example, batch normalization aims to improve training by normalizing the inputs to a layer across a batch – allowing the network to focus only on the variations in training data rather than the potentially overwhelming average signal (Ioffe & Szegedy, 2015). Layer normalization is a technique used in training recurrent neural networks and removes the dependence on batch size (Ba et al., 2016). These normalizations techniques are more complex than what we suggest. Our modifications are not aiming to supersede these techniques in the Dense Associative Memory but simply improve network stability and practicality on an implementation level. Moreover, our suggestions do not exclude the possibility of using these other normalization techniques.

Networks related to the Dense Associative Memory have employed some normalization techniques in a similar manner to our work. Perhaps most closely related is the continuous, attention-like Hopfield network (Ramsauer et al., 2021) which has shown promising results in the realm of transformer architectures. Ramsauer et al. operate over a slightly different domain; spherical vectors rather than bipolar vectors. While the vector magnitude is still constant, the network has changed rather significantly from the one introduced by Krotov and Hopfield, which may slightly change the arguments we make below. However, we note that no analysis of the network stability in relation to floating point accuracy is made, and the remainder of our modifications are not applied (e.g. shifting scaling factors inside the interaction function), which our work expands on considerably. Further works have applied a similar normalization, albeit without noting why it is useful for network stability (Millidge et al., 2022; Liang et al., 2022; Alonso & Krichmar, 2024). Literature on Dense Associative Memory applications and derivatives discuss normalization either in a separate context or only tangentially. Extensive work has been done on contrastive normalization (a biologically plausible explanation of network behavior) in the Dense Associative Memory and its relation to the restricted Boltzmann machine (Krotov & Hopfield, 2021). Other research employs advanced normalization techniques, including some we discuss above such as layer normalization, by treating the Dense Associative Memory as a deep recurrent network (Seidl et al., 2021). Again, these works do not consider shifting the scaling factors within the interaction function.

## 3 FORMALIZATION OF THE HOPFIELD NETWORK AND DENSE ASSOCIATIVE MEMORY

The Hopfield network defines a weight matrix based on the Hebbian of the learned states $\xi$, indexed by $\mu$:

$$W_{ji} = \sum_{\mu} \xi_j^{\mu} \xi_i^{\mu} \tag{3}$$

The update dynamics for a probe state $\xi$ are defined by the sign of the energy function, with updates being applied asynchronously across neurons:

$$\xi_i^{(t+1)} = \text{sign}\left( \sum_j W_{ji} \xi_j^{(t)} \right), \tag{4}$$

where sign is the sign function, or hardlimiting activation function:

$$\text{sign}(x) = \begin{cases} 1 & \text{if } x \geq 0, \\ -1 & \text{if } x < 0. \end{cases} \tag{5}$$

The Dense Associative Memory has significantly different learning rules and update dynamics compared to the Hopfield network, as well as major architectural changes, such as using a set of memory vectors $\zeta$ instead

of a weight matrix $W$. The Dense Associative Memory also does away with a simple energy function and instead uses the sign of the difference of energies. The unmodified update rule has the form:

$$\xi_i^{(t+1)} = \text{sign}\left[\sum_{\mu}\left(F_n(\zeta_i^{\mu} + \sum_{j\neq i}\zeta_j^{\mu}\xi_j^{(t)}) - F_n(-\zeta_i^{\mu} + \sum_{j\neq i}\zeta_j^{\mu}\xi_j^{(t)})\right)\right] \tag{6}$$

Where $F_n$ is the interaction function, parameterized by interaction vertex $n$. The interaction vertex controls how steep the interaction function is. Common interaction functions include the polynomial

$$F_n(x) = x^n, \tag{7}$$

rectified polynomial

$$F_n(x) = \begin{cases} x^n & \text{if } x \geq 0, \\ 0 & \text{if } x < 0, \end{cases} \tag{8}$$

or leaky rectified polynomial

$$F_n(x, \epsilon) = \begin{cases} x^n & \text{if } x \geq 0, \\ -\epsilon x & \text{if } x < 0. \end{cases} \tag{9}$$

The Hopfield network behavior is recovered when using the polynomial interaction function in Equation 7 and $n = 2$ (Krotov & Hopfield, 2016; Demircigil et al., 2017). Increasing the interaction vertex allows memory vectors to affect only very similar probe vectors, decreasing interference with other memory vectors.

The Hopfield network requires only the energy calculation of the current state for updates (Equation 4), while the Dense Associative Memory requires the calculation of the energy for the current state when neuron $i$ is clamped on (value 1) and clamped off (value $-1$). This is more computationally expensive but allows for updating when the interaction vertex is larger than 2 and the usual arguments for update convergence in the Hopfield network fail (Hopfield, 1982; Hopfield & Tank, 1985).

Instead of a weight matrix, the Dense Associative Memory uses a set of memory vectors, clamped to have values between $-1$ and 1, but not necessarily corresponding to the learned states. Instead, the learned states are used to update the memory vectors in a gradient descent. The unmodified loss function used in the gradient descent is based on the update rule in Equation 6:

$$\mathcal{L} = \sum_a \sum_i (\xi_{a,i} - C_{a,i})^{2m}$$

$$C_{a,i} = \tanh\left[\beta\sum_{\mu}\left(F_n(\zeta_i^{\mu} + \sum_{j\neq i}\zeta_j^{\mu}\xi_{a,j}^{(t)}) - F_n(-\zeta_i^{\mu} + \sum_{j\neq i}\zeta_j^{\mu}\xi_{a,j}^{(t)})\right)\right] \tag{10}$$

Where $a$ indexes over the learned states, and $i$ indexes over the neurons. The new parameters $m$ and $\beta$ control the learning process. The error exponent $m$ emphasizes larger errors, and the inverse temperature $\beta$ scales the argument of the $\tanh$ function, avoiding vanishing gradients as the argument grows in magnitude. Krotov & Hopfield (2016) found the error exponent can help train higher interaction vertex networks, and suggest $\beta = \frac{1}{T^n}$, with hyperparameter $T$ representing the network temperature.

Inspecting the order of calculations in Equation 6 and 10: first the "similarity score" between a learned state and a memory vector is calculated $\pm\zeta_i^{\mu} + \sum_{j\neq i}\zeta_j^{\mu}\xi_j^{(t)}$, effectively the dot product between two binary vectors of length equal to the network dimension $N$. Next, this similarity score is passed into the interaction function, which will typically have a polynomial-like form such as in Equation 7 or 8. If the interaction vertex $n$ is large the memory vectors become prototypes of the learned states (Krotov & Hopfield, 2016),

hence the similarity scores will approach the bound for the dot product of two binary vectors, $N$. We may have to calculate a truly massive number as an intermediate value. For example, $N = 10^4$ and $n = 30$ will result in an intermediate value of $10^{120}$. Single precision floating point numbers ("floats") have a maximum value of around $10^{38}$, while double precision floating point numbers ("doubles") have a maximum value of around $10^{308}$. In our example, we are already incapable of even storing the intermediate value in a float, and it would not require increasing the network dimension or interaction vertex considerably to break a double. Furthermore, the precision of these data types decreases as we approach the limits, potentially leading to numerical instabilities during training or updating. Even in the update rule (Equation 6) where only the sign of the result is relevant, a floating point overflow renders the calculation unusable.

We propose a slight modification to the implementation of the Dense Associative Memory. Normalizing the similarity score by the network dimension $N$ bounds the magnitude of the result to 1 rather than $N$. Additionally, we propose pulling the scaling factor $\beta$ inside the interaction function, so we can appropriately scale the value before any imprecision is introduced by a large exponentiation as well as controlling the gradient, making the network more robust. We show these modifications are equivalent to the original Dense Associative Memory specification in Section 4. In Section 5 we also show by experimentation that these modifications make the network temperature independent of the interaction vertex. This makes working with the Dense Associative Memory more practical, as it avoids large hyperparameter searches when slightly altering the interaction vertex.

## 4 Modification and Consistency with Original

Our modifications attempt to rectify the floating point issues by scaling the similarity scores *before* applying the exponentiation of the interaction function. To justify our modifications we must show that the scaling has no effect on the properties of the Dense Associative Memory in both learning and updating. For the update rule, we will show the sign of the argument to the hardlimiting function in Equation 6 is not affected as we introduce a scaling factor and move it within the interaction function. For learning, we will make a similar argument using Equation 10.

### 4.1 Homogeneity of the Interaction Function

In parts of our proof on the modification of the Dense Associative Memory we require the interaction function to have a particular form. We require the sign of the difference of two functions remain constant even when a scaling factor is applied inside those functions; $f(x) - f(y) = f(\alpha x) - f(\alpha y) \quad \forall a > 0$. A stronger property (that is much easier to prove) is that of homogeneity:

$$f(\alpha x) = \alpha^k f(x) \qquad \forall \alpha > 0 \tag{11}$$

with the exponent $k$ known as the degree of homogeneity.

**Lemma 4.1.1.** *The polynomial interaction function (Equation 7) is homogenous.*

*Proof.*

$$\begin{aligned} F_n(\alpha x) &= (\alpha x)^n \\ &= \alpha^n x^n \\ &= \alpha^n F_n(x) \end{aligned}$$

Hence, the polynomial interaction function is homogenous, with degree of homogeneity equal to the interaction vertex $n$. $\qquad\square$

**Lemma 4.1.2.** *The rectified polynomial interaction function (Equation 8) is homogenous.*

*Proof.*

$$F_n\left(\alpha x\right) = \begin{cases} (\alpha x)^n & \text{if } (\alpha x) \geq 0 \\ 0 & \text{if } (\alpha x) < 0 \end{cases} = \begin{cases} \alpha^n x^n & \text{if } (\alpha x) \geq 0 \\ 0 & \text{if } (\alpha x) < 0, \end{cases}$$

$$= \alpha^n \begin{cases} x^n & \text{if } x \geq 0 \\ 0 & \text{if } x < 0, \end{cases} = \alpha^n F_n(x)$$

Note that the sign of $x$ is unchanged by scaling by $\alpha > 0$, so we can change the conditions on the limits as we have. Hence, the rectified polynomial interaction function is homogenous, with degree of homogeneity equal to the interaction vertex $n$. □

### 4.1.1 ON COMMON NONHOMOGENOUS INTERACTION FUNCTIONS

The leaky rectified polynomial interaction function (Equation 9) is common in literature, alongside Equation 7 and 8. However, the leaky rectified polynomial is not homogenous. Empirically, we find it still behaves well under our modifications.

The Dense Associative Memory has been generalized further using an exponential interaction function (Demircigil et al., 2017). Another modification of the exponential interaction function has been used to allow for continuous states and an exponential capacity (Ramsauer et al., 2021). This interaction function has been analyzed in depth and linked to the attention mechanism in transformer architectures (Vaswani et al., 2017).

$$F\left(x\right) = e^x, \tag{12}$$

The exponential interaction function is not homogenous. However, we can analyze the exponential function specifically and relax the homogeneity constraint to show our modifications will not affect networks with exponential interaction functions. In particular, we need only show the sign of the difference between two exponentials is unaffected:

$$\text{sign}\left[\alpha\left(e^x - e^y\right)\right] = \text{sign}\left[\alpha e^x - \alpha e^y\right]$$
$$= \text{sign}\left[e^{\log(\alpha)}e^x - e^{\log(\alpha)}e^y\right]$$
$$= \text{sign}\left[e^{\log(\alpha)x} - e^{\log(\alpha)y}\right].$$

Therefore, our modifications will not affect the properties of the Dense Associative Memory when using the exponential interaction function, as we are still free to choose any scaling factor $\alpha$.

### 4.2 UPDATE IN THE DENSE ASSOCIATIVE MEMORY

We start with the right-hand side of Equation 6, introducing an arbitrary constant $\alpha > 0$. We will then show this has no effect on the sign of the result, and we are free to choose $\alpha = \frac{1}{N}$ to normalize the similarity scores by the network dimension.

**Theorem 4.2.1.** *The Dense Associative Memory, equipped with a homogenous interaction function, has unchanged update dynamics (Equation 6) when applying a scaling factor $\alpha > 0$ to similarity calculations inside the interaction function. That is:*

$$\text{sign}\left[\sum_{\mu}\left(F_n(\zeta_i^\mu + \sum_{j\neq i}\zeta_j^\mu\xi_j^{(t)}) - F_n(-\zeta_i^\mu + \sum_{j\neq i}\zeta_j^\mu\xi_j^{(t)})\right)\right]$$

$$= \text{sign}\left[\sum_{\mu}\left(F_n(\alpha(\zeta_i^\mu + \sum_{j\neq i}\zeta_j^\mu\xi_j^{(t)})) - F_n(\alpha(-\zeta_i^\mu + \sum_{j\neq i}\zeta_j^\mu\xi_j^{(t)}))\right)\right].$$

*Proof.* The sign of any real number is unaffected by scaling factor $\alpha > 0$:

$$\text{sign}\left[\alpha\sum_{\mu}\left(F_n(\zeta_i^\mu + \sum_{j\neq i}\zeta_j^\mu\xi_j^{(t)}) - F_n(-\zeta_i^\mu + \sum_{j\neq i}\zeta_j^\mu\xi_j^{(t)})\right)\right]$$

$$= \text{sign}\left[\sum_{\mu}\left(\alpha F_n(\zeta_i^\mu + \sum_{j\neq i}\zeta_j^\mu\xi_j^{(t)}) - \alpha F_n(-\zeta_i^\mu + \sum_{j\neq i}\zeta_j^\mu\xi_j^{(t)})\right)\right]$$

$$= \text{sign}\left[\sum_{\mu}\left(F_n(\alpha'(\zeta_i^\mu + \sum_{j\neq i}\zeta_j^\mu\xi_j^{(t)})) - F_n(\alpha'(-\zeta_i^\mu + \sum_{j\neq i}\zeta_j^\mu\xi_j^{(t)}))\right)\right]$$

Using the assertion that $F_n$ is homogenous in the last step. Since the scaled factor $\alpha'$ is still arbitrary, we are free to select any (positive) value we like without changing the result. □

Hence, our modified update rule in Equation 1 will give the same behavior as the original update rule in Equation 6. As discussed, we suggest choosing $\alpha = \frac{1}{N}$, the inverse of the network dimension, such that the similarity scores are normalized between $-1$ and $1$. This nicely avoids floating point overflow.

### 4.3 LEARNING IN THE DENSE ASSOCIATIVE MEMORY

Reasoning about the learning rule (Equation 10) is slightly trickier than the update rule. We must ensure the network learning remains consistent with the unmodified network as to not throw away much of the theoretical work on, say, the capacity of the Dense Associative Memory. Furthermore, there is already a scaling factor $\beta$ present; adding a second, independent hyperparameter would only complicate the network further. We will show that we can pull the existing scaling factor within the interaction function and keep its intended action of shifting the argument of the $\tanh$ function, and hence that we can achieve the same calculation as the original network. The argument here is largely the same as in Section 4.2.

**Theorem 4.3.1.** *The Dense Associative Memory, equipped with a homogenous interaction function, has unchanged learning behavior (Equation 10) when moving the scaling factor $\beta$ inside the interaction function evaluations, up to adjusting the scaling factor. That is:*

$$\tanh\left[\beta'\sum_{\mu}\left(F_n(\zeta_i^\mu + \sum_{j\neq i}\zeta_j^\mu\xi_j^{(t)}) - F_n(-\zeta_i^\mu + \sum_{j\neq i}\zeta_j^\mu\xi_j^{(t)})\right)\right]$$

$$= \tanh\left[\sum_{\mu}\left(F_n(\beta(\zeta_i^\mu + \sum_{j\neq i}\zeta_j^\mu\xi_j^{(t)})) - F_n(\beta(-\zeta_i^\mu + \sum_{j\neq i}\zeta_j^\mu\xi_j^{(t)}))\right)\right].$$

*Proof.* Equation 10 defines a loss function over which a gradient descent is applied to update the memory vectors $\zeta$. To show this gradient descent is unchanged by moving the scaling factor $\beta$ inside the interaction function evaluations, we focus on the predicted neuron value in Equation 10 and apply the same algebra as in Theorem 4.2.1 to take the scaling factor $\beta'$ inside the interaction function. Note that this also requires the homogeneity of the interaction function, and may alter the value the scaling factor $\beta$, but will ensure the argument to the $\tanh$ (and hence the gradient) remains the same. The exact gradient expression is eschewed here but remains unchanged from the original. □

Therefore, our modified learning rule in Equation 2 is a suitable replacement for the original in Equation 10. Krotov and Hopfield suggest a value of $\beta = \frac{1}{T^n}$. We suggest a modified value of $\beta = \frac{1}{NT}$, as to normalize the similarity scores once again.

## 5 HYPERPARAMETER TUNING

The original Dense Associative Memory suffers from very strict hyperparameter requirements. Changing the value of the interaction vertex significantly changes the optimal hyperparameters for training. We find that our modifications — particularly, normalizing the similarity scores in the learning rule — remove the dependence on the interaction vertex.

We focus on the most important hyperparameters for learning: the initial learning rate and temperature. Other hyperparameters were tuned but did not display behavior as dramatic as we present here. We use a learning rate decay of $0.999$ per epoch, a momentum of $0.0$, and an error exponent of $m = 1.0$. We found similar results using a decay rate of $1.0$ and higher values for momentum. We also found we did not require changing the error exponent $m$, which Krotov & Hopfield (2016) found to be useful in learning higher interaction vertices. This perhaps indicates we can remove this hyperparameter and simplify the network.

The network is trained on 20 randomly generated bipolar vectors of dimension 100. Even for the lowest interaction vertex $n = 2$ this task is perfectly learnable. Larger dimensions and other dataset sizes were tested with similar results. After training, we probe the network with the learned states; if the probes move only a small distance from the learned states, the network operates as an acceptable associative memory. We measure the average distance from the final, stable states to the learned states, for which a lower value is better. We repeat the experiment five times for each combination of hyperparameters, of which select interaction vertices are shown in Figure 1. All of our results can be found in Appendix A. In particular, Appendix A.3 shows our results for interaction vertices up to $n = 100$; far above any interaction vertices documented in other literature. Note that for the unmodified network have avoided floating point overflow by engineering our experiments to remain within the bounds of a double. The performance degradation seen at larger interaction vertices is not due to floating point overflow.

In the original network, the optimal hyperparameter region shifts considerably with the interaction vertex. At even modest interaction vertices we find the optimal region is fleeting enough to not appear in our grid search. It is tempting to claim that a finer grid search may reveal the region to persist. However, inspection of Figure 1e shows that not only has the optimal region vanished at this granularity, but the same region now has high distance measure. Even if the optimal region exists and is very small, it is apparently surrounded by an increasingly suboptimal region. This is troublesome and makes working with the network difficult, especially when altering the interaction vertex even slightly.

For our modified implementation of the Dense Associative Memory, we have shifted the scale of the inverse temperature as discussed in Section 4. We find the optimal region shifts slightly for small interaction vertices, but unlike the original network we find the region stabilizes and remains substantial for large interaction vertices. Most notably, we find that the optimal region's position remains stable and size remains large across many values of the interaction vertex for the same network dimension.

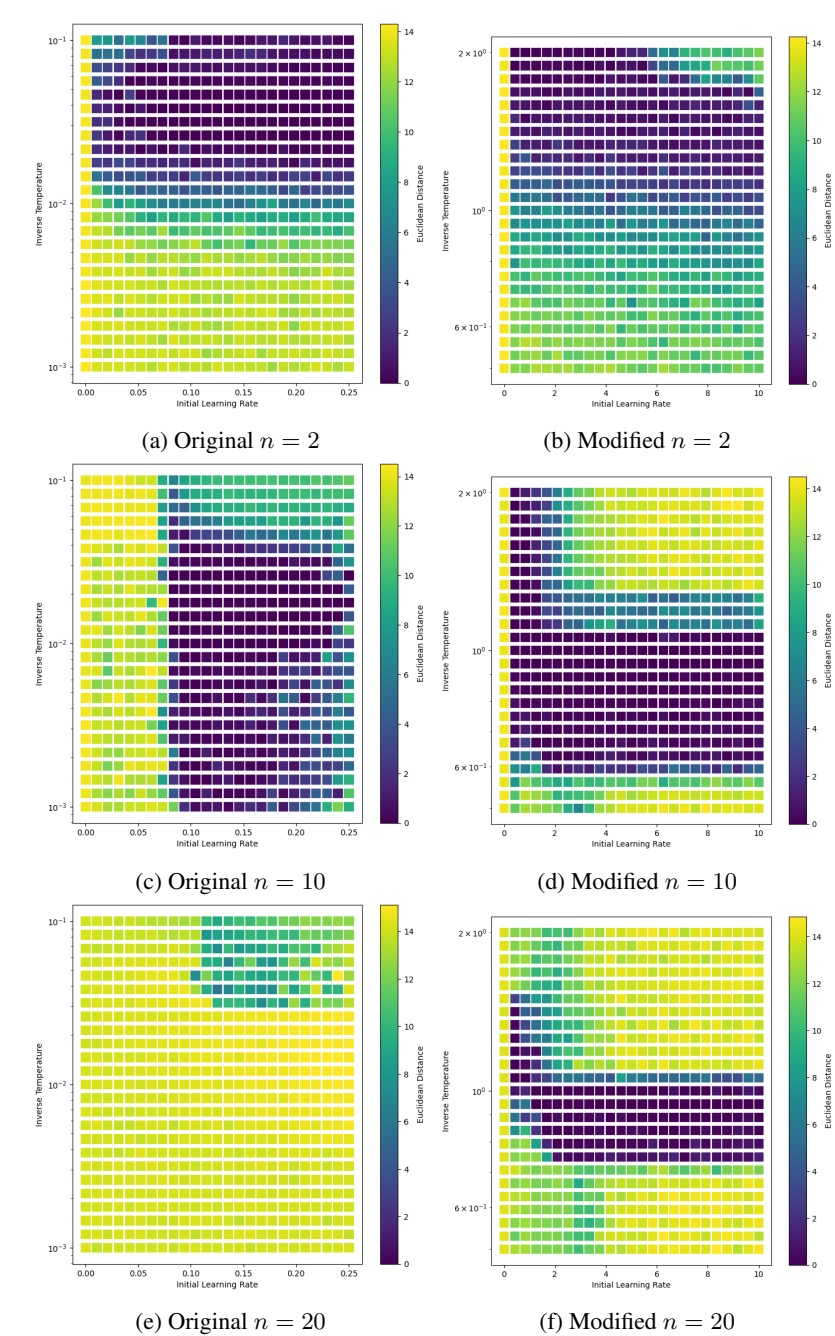

Figure 1: Select hyperparameter searches for an autoassociative memory task, measuring the Euclidean distance between learned states and relaxed states. The left column shows results from the original, unmodified network, while the right column shows results from our modified network, and rows showing results from various interaction vertices. Smaller distances correspond to better recall and hence better a better associative memory.

## 5.1 HYPERPARAMETER SELECTION IN CLASSIFICATION TASKS

We have focused on the Dense Associative Memory as an autoassociative memory, where all neurons are updated at each step and may be updated numerous times until the state reaches stability. Another, perhaps more popular use case of the network in current literature is as a classifier. By splitting the memory vectors into two parts – a section for input data and a section for classes as logits – the network can be run as a classifier by only updating the classification neurons, and only updating those neurons once (Krotov & Hopfield, 2016).

We conduct a similar hyperparameter search over both the original and modified implementation of the Dense Associative Memory for a classification as we did for the autoassociative tasks in Section 5. Specifically, we trained a Dense Associative Memory to classify the MNIST dataset and measured the validation F1 score for a validation split of 20%. We found that the optimal hyperparameter region in classification tasks was of roughly the same shape, size, and relative location between both the original and modified implementations for each respective interaction vertex. This indicates that our modifications have preserved the hyperparameter stability in classification based tasks, but not improved as seen in Section 5. However, we also observe that the optimal inverse temperature in our modified implementation to be consistently around $\beta \approx 1$, meaning we have a better idea of where to search for optimal hyperparameters. While not as significant a result as in autoassociative tasks, this result is still useful in working with the Dense Associative Memory as the location of the optimal hyperparameter region is consistent across different datasets and tasks. Our full results of these experiments can be found in Appendix B.

## 6 CONCLUSION

In this work, we have investigated the technical details of the Dense Associative Memory and its implementation. We note that the original network specification leads to floating point imprecision and overflow when calculating intermediate values for both update and learning. We provide details on when this imprecision occurs and show the conditions are more likely when the interaction vertex is large based on the feature-to-prototype transition of the memory vectors (Krotov & Hopfield, 2016). We propose a modification to the network implementation that prevents the floating point issues. We prove our modifications do not alter the network properties, such as the capacity and autoassociative nature. Our proof relies on the interaction function being homogenous, however this property is stronger than is required, and we find empirically that some nonhomogenous functions also give well-behaved Dense Associative Memories. We then show our modified network has optimal hyperparameter regions that do not shift based on the choice of interaction vertex for purely autoassociative tasks. For classification like tasks, such as MNIST classification, our modifications do not appear to radically improve the optimal hyperparameter region but rather shift the region to a common location that makes tuning the network easier. Our modifications greatly simplify working with the Dense Associative Memory, as experiments on a dataset do not need to search across a potentially large hyperparameter space for each change in the interaction vertex. We also find several hyperparameters do not need tuning in our experiments, hinting at a potentially simpler network that is easier to tune and interpret.

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

# A    FULL RESULTS OF HYPERPARAMETER SEARCHES

## A.1    ORIGINAL NETWORK, DIMENSION 100

These results are from the original network, have dimension 100, and train on 20 learned states.

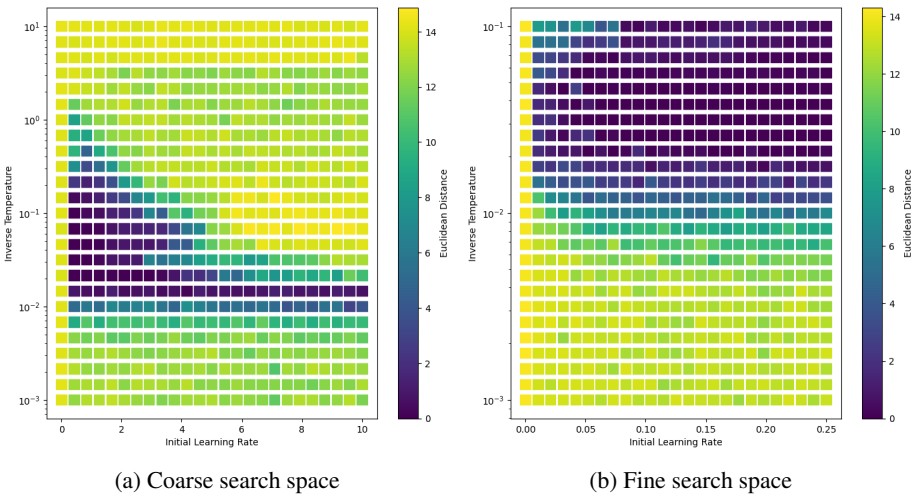

(a) Coarse search space

(b) Fine search space

Figure 2: Hyperparameter search space for $n = 2$

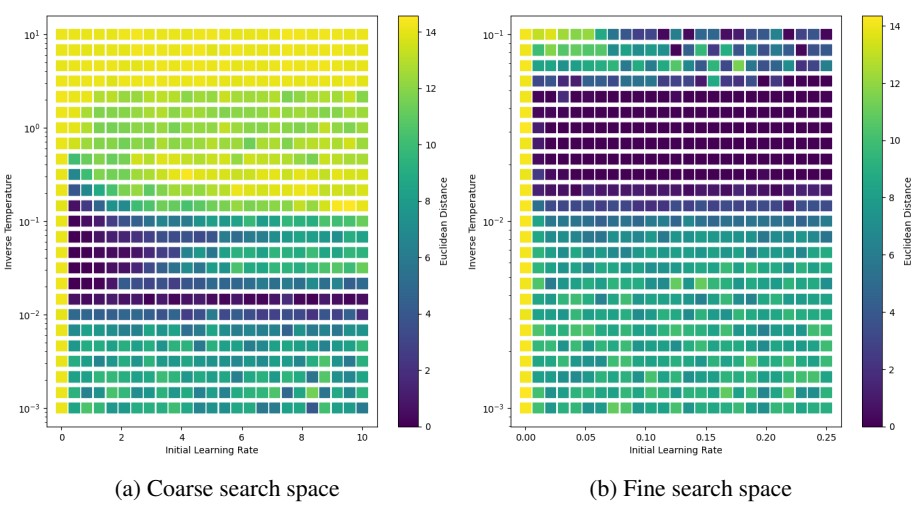

(a) Coarse search space

(b) Fine search space

Figure 3: Hyperparameter search space for $n = 3$

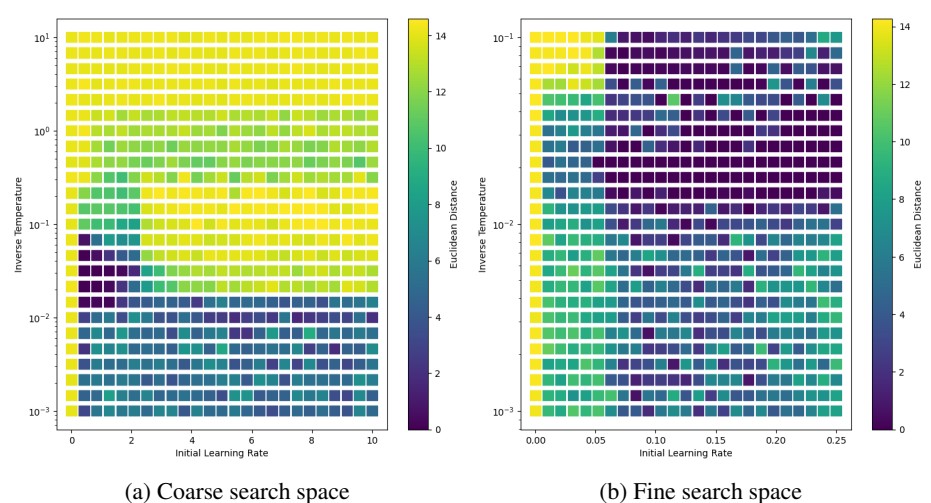

(a) Coarse search space        (b) Fine search space

Figure 4: Hyperparameter search space for $n = 5$

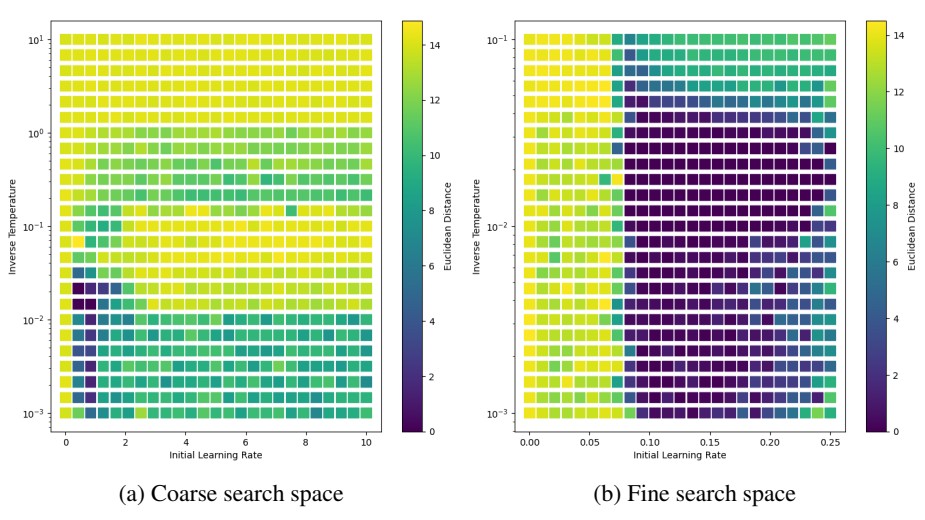

(a) Coarse search space        (b) Fine search space

Figure 5: Hyperparameter search space for $n = 10$

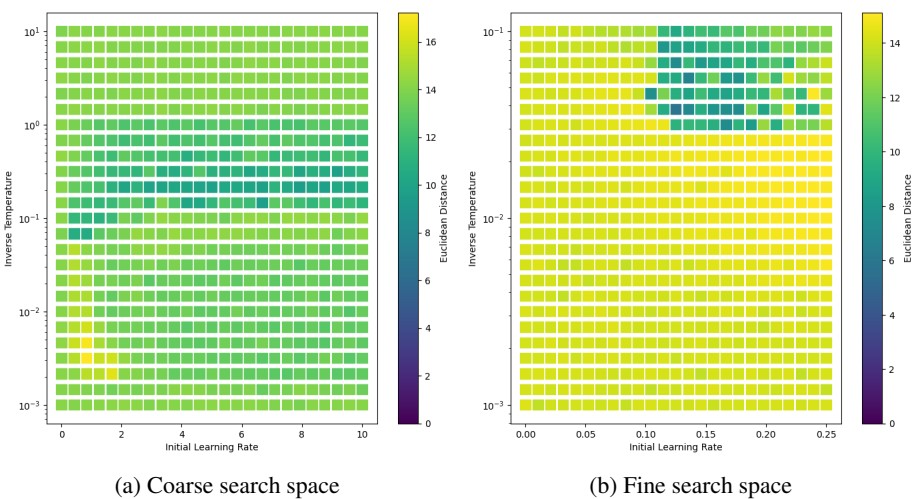

(a) Coarse search space        (b) Fine search space

Figure 6: Hyperparameter search space for $n = 20$

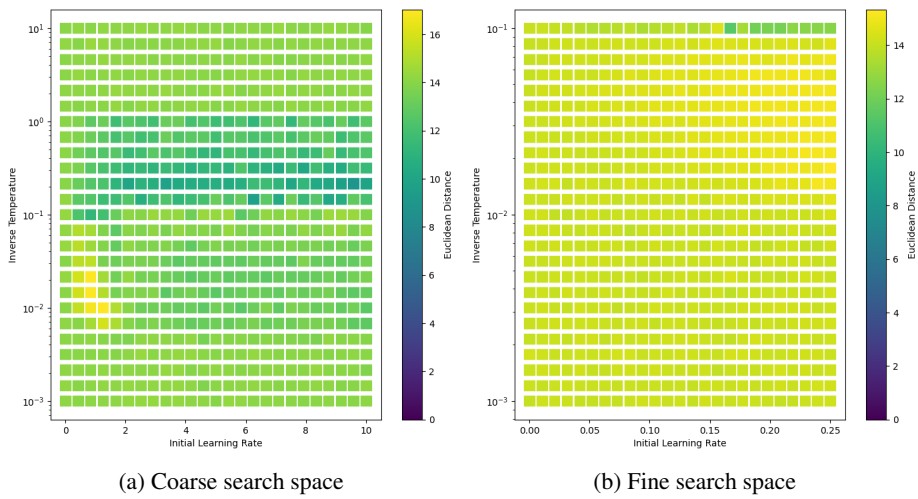

(a) Coarse search space        (b) Fine search space

Figure 7: Hyperparameter search space for $n = 30$

## A.2 MODIFIED NETWORK, DIMENSION 100

These results are from our modified network, have dimension 100, and train on 20 learned states.

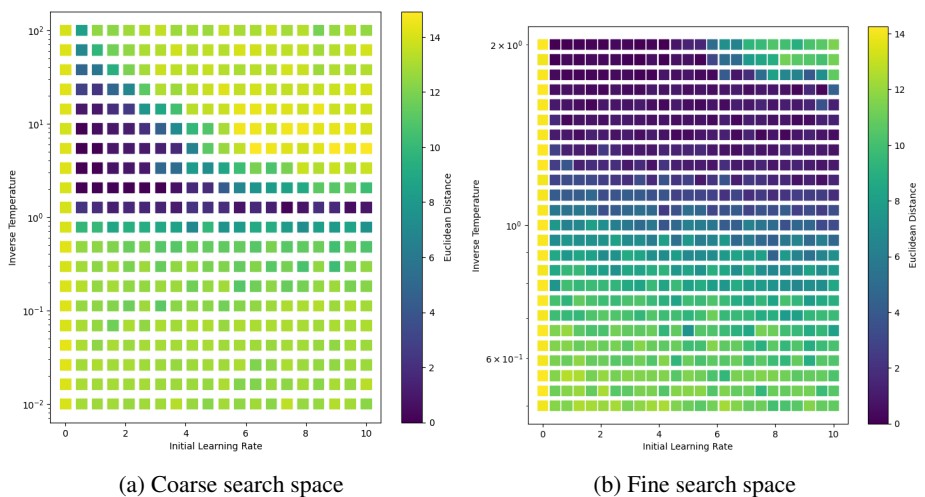

(a) Coarse search space

(b) Fine search space

Figure 8: Hyperparameter search space for $n = 2$

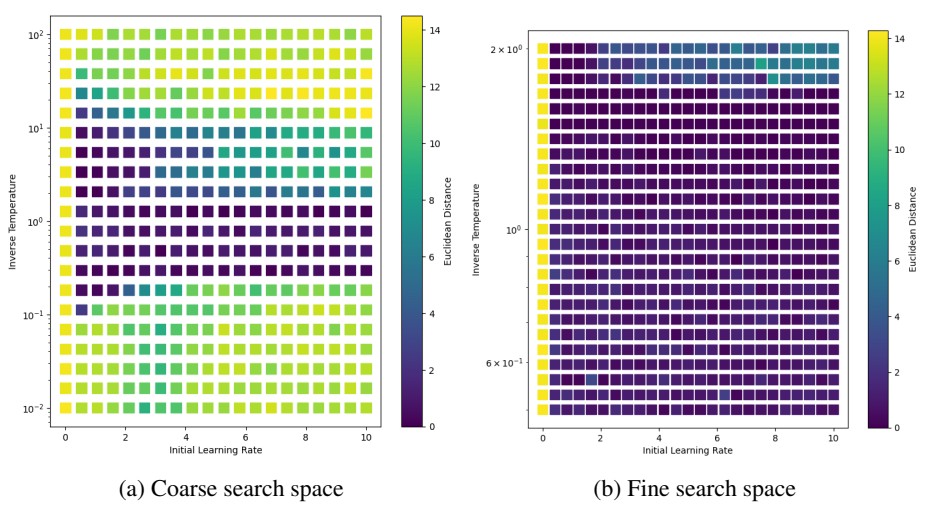

(a) Coarse search space

(b) Fine search space

Figure 9: Hyperparameter search space for $n = 3$

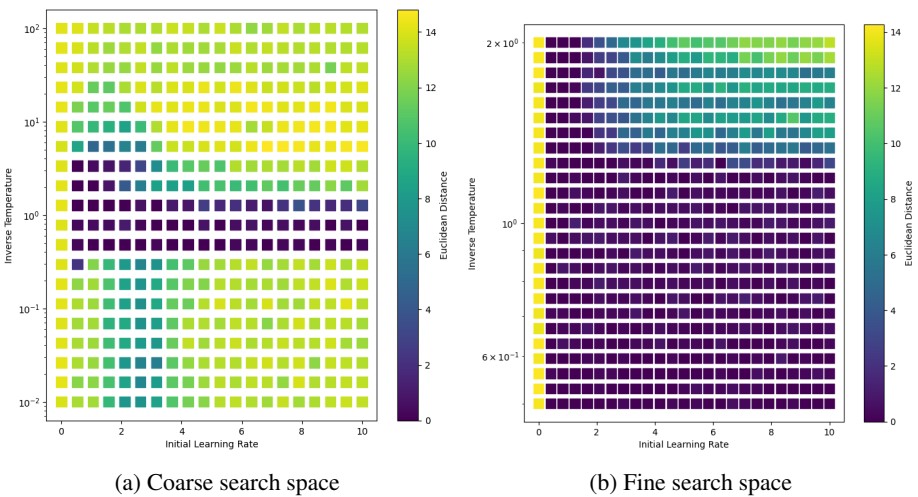

(a) Coarse search space       (b) Fine search space

Figure 10: Hyperparameter search space for $n = 5$

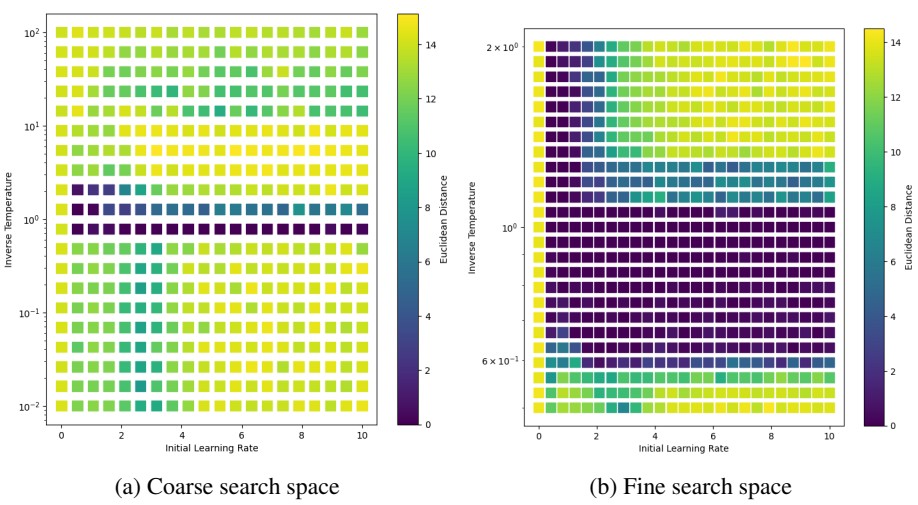

(a) Coarse search space       (b) Fine search space

Figure 11: Hyperparameter search space for $n = 10$

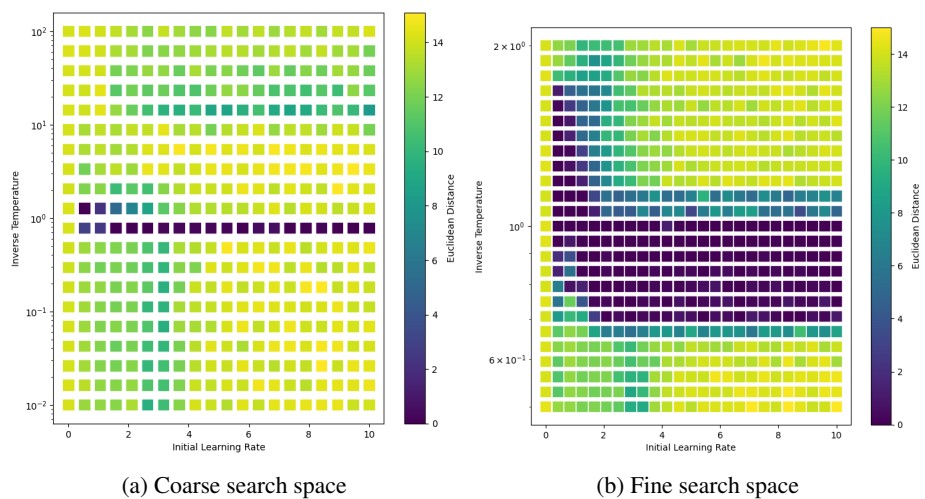

(a) Coarse search space      (b) Fine search space

Figure 12: Hyperparameter search space for $n = 15$

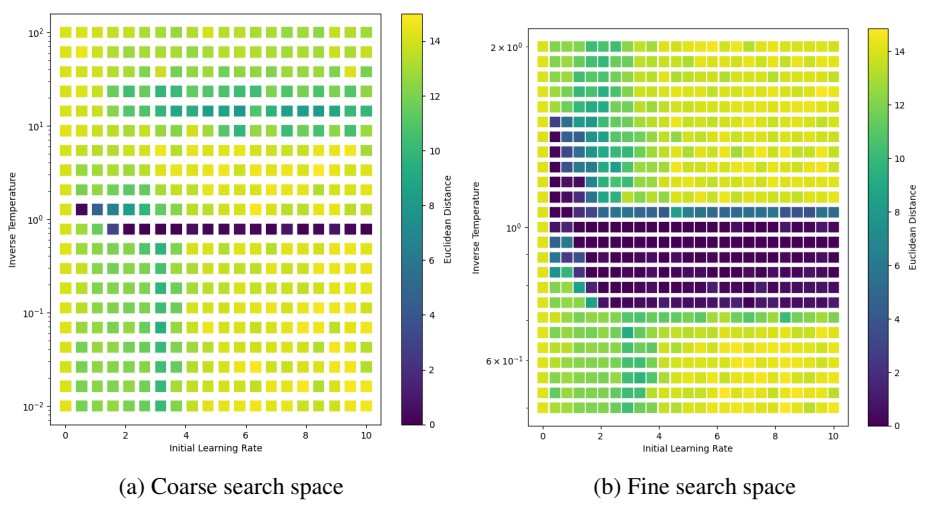

(a) Coarse search space      (b) Fine search space

Figure 13: Hyperparameter search space for $n = 20$

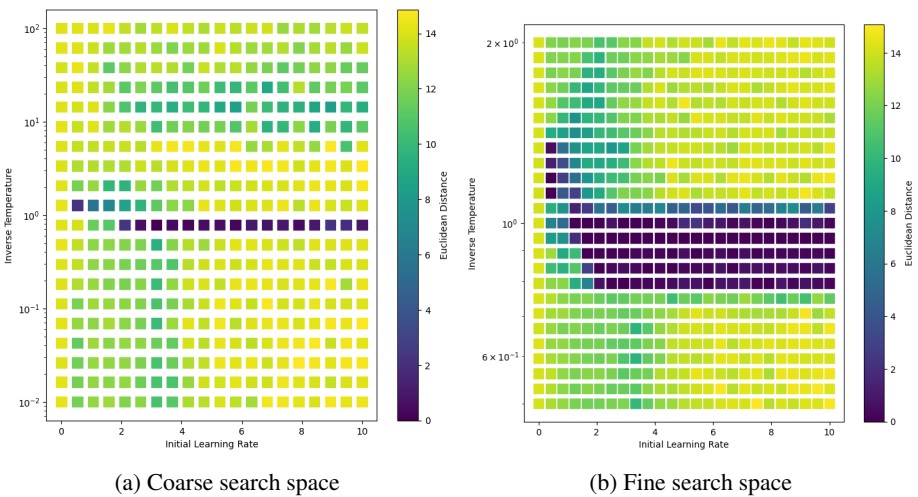

(a) Coarse search space        (b) Fine search space

Figure 14: Hyperparameter search space for $n = 25$

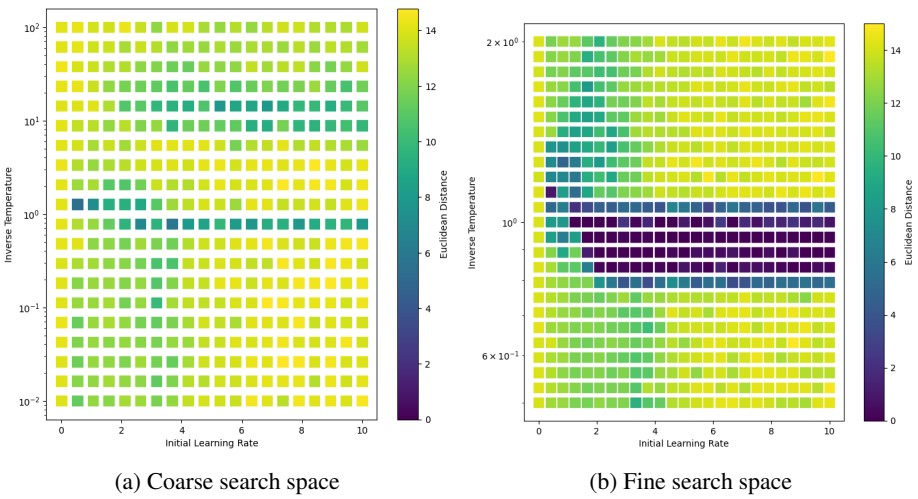

(a) Coarse search space        (b) Fine search space

Figure 15: Hyperparameter search space for $n = 30$

### A.3  MODIFIED NETWORK, DIMENSION 100, LARGE INTERACTION VERTEX

These results continue with the same network and setup from Appendix A.2 but with much larger interaction vertices than were possible with the original network. We also present only the tight grid search results, as the coarse grid search did not capture the optimal region well.

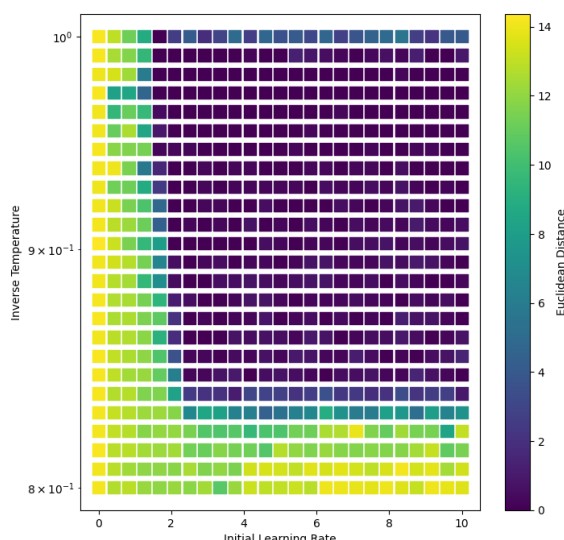

Figure 16: Hyperparameter search space for $n = 40$

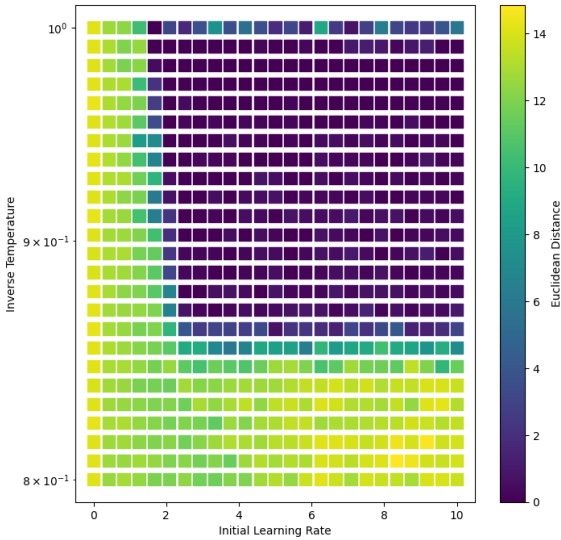

Figure 17: Hyperparameter search space for $n = 50$

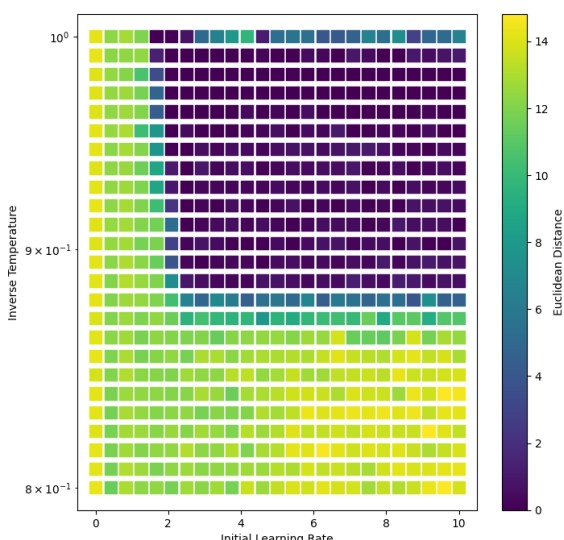

Figure 18: Hyperparameter search space for $n = 60$

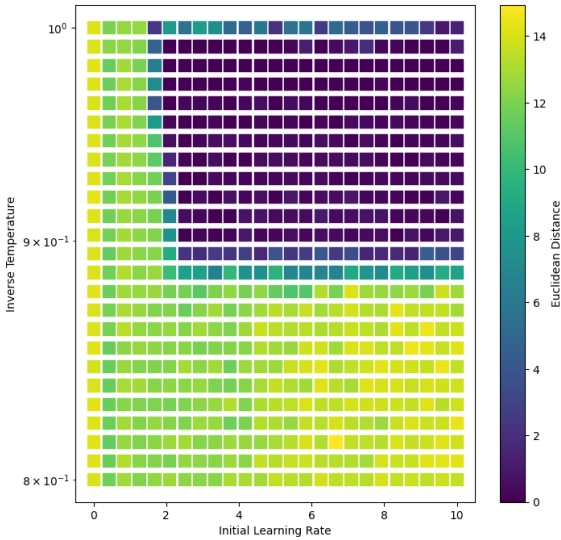

Figure 19: Hyperparameter search space for $n = 70$

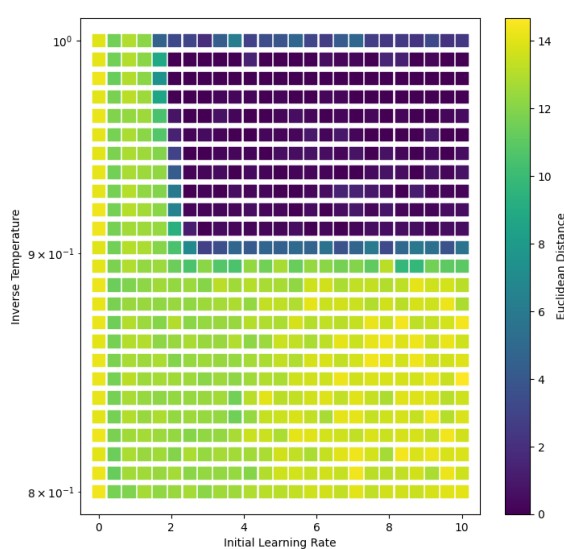

Figure 20: Hyperparameter search space for $n = 80$

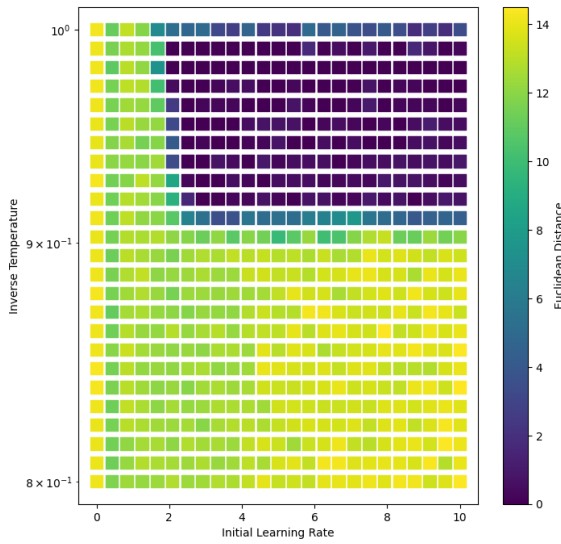

Figure 21: Hyperparameter search space for $n = 90$

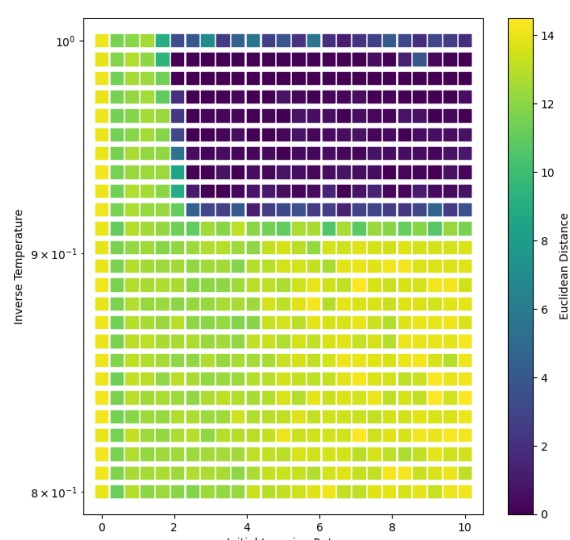

Figure 22: Hyperparameter search space for $n = 100$

## A.4 MODIFIED NETWORK, DIMENSION 250

These results are from our modified network, have dimension 250, and train on 30 learned states.

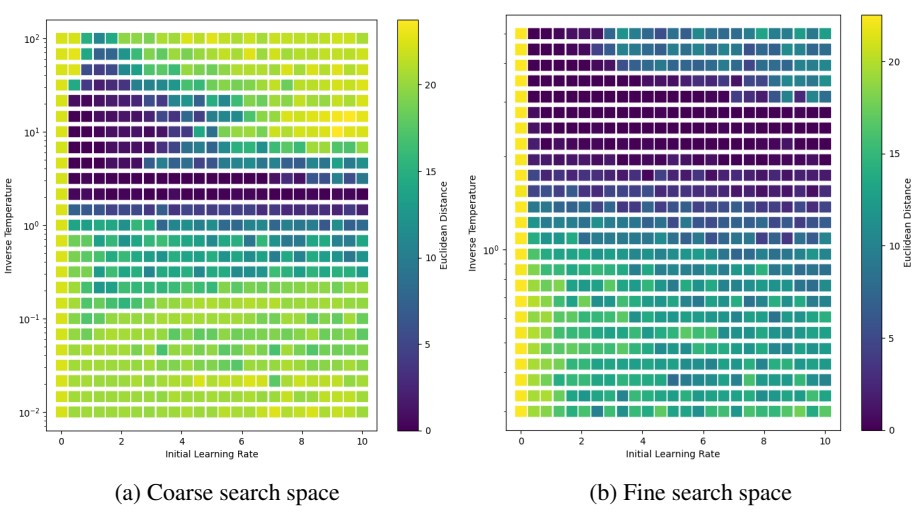

(a) Coarse search space        (b) Fine search space

Figure 23: Hyperparameter search space for $n = 2$

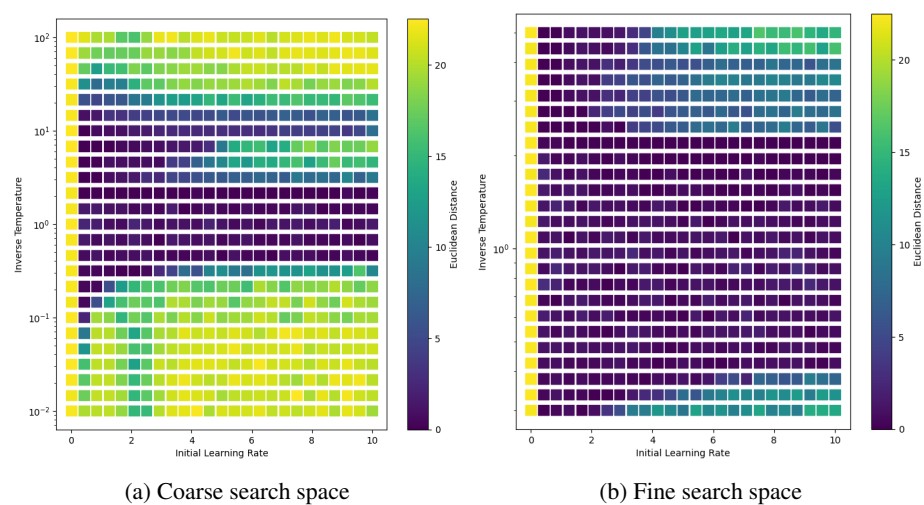

(a) Coarse search space

(b) Fine search space

Figure 24: Hyperparameter search space for $n = 3$

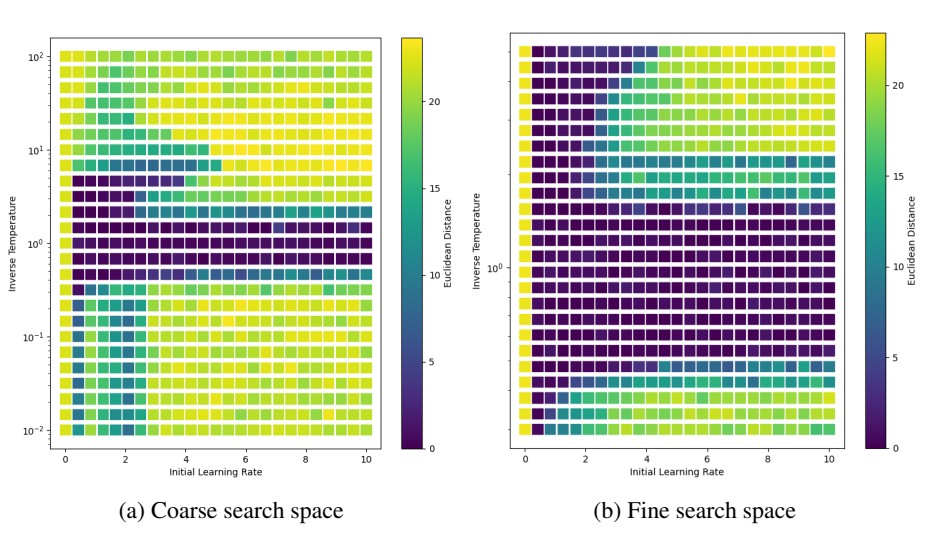

(a) Coarse search space

(b) Fine search space

Figure 25: Hyperparameter search space for $n = 5$

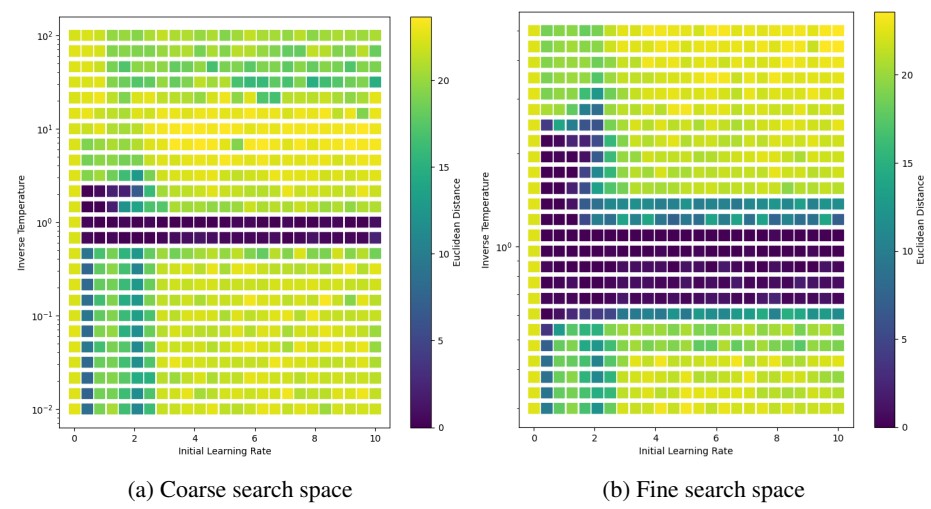

(a) Coarse search space     (b) Fine search space

Figure 26: Hyperparameter search space for $n = 10$

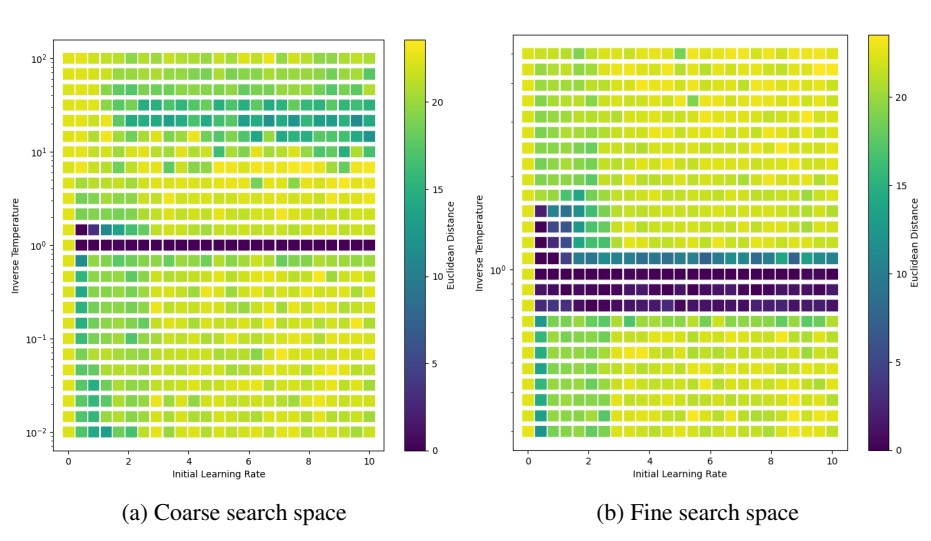

(a) Coarse search space     (b) Fine search space

Figure 27: Hyperparameter search space for $n = 20$

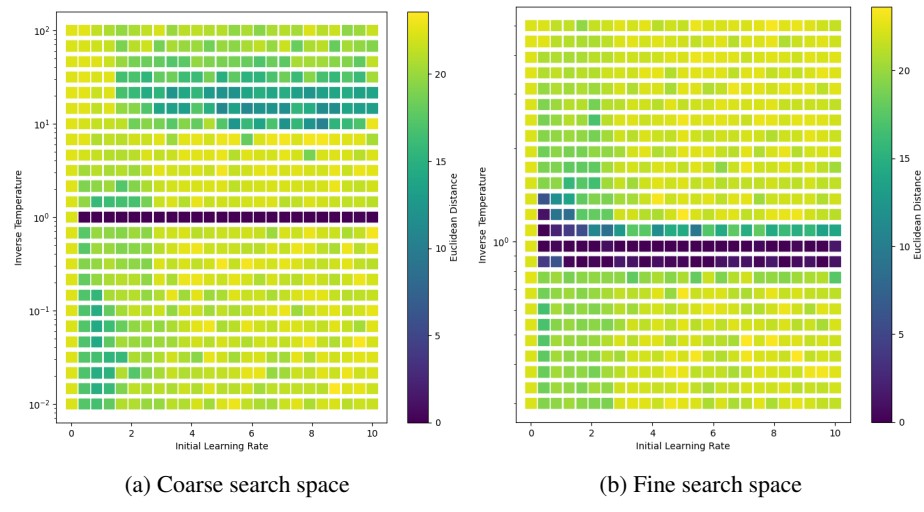

(a) Coarse search space            (b) Fine search space

Figure 28: Hyperparameter search space for $n = 30$

## B  HYPERPARAMETER SEARCH OVER MNIST TASK

In our results below, we have trained the Dense Associative Memory on the MNIST dataset and note the validation F1 score across hyperparameter space, meaning for the following results we are aiming for larger values, not smaller as above. We have used the autoassociative memory model, rather than the feed-forward equivalent shown by Krotov & Hopfield (2016). This means we have *not* explicitly ignored the effects of the classification neurons on one another, as is done in constructing the feed-forward equivalent, although the effect is likely negligible. Note that we have significantly different scales for the original and modified network's values of $\beta$, which is not seen in the previous results. We believe this is due to leaving some memory weights unclamped, as well as only updating a small number of neurons as required for classification. Notably, our range of $\beta$ for the original network matches the range found by (Krotov & Hopfield, 2016). In all experiments we trained the network for 500 epochs with 256 memory vectors.

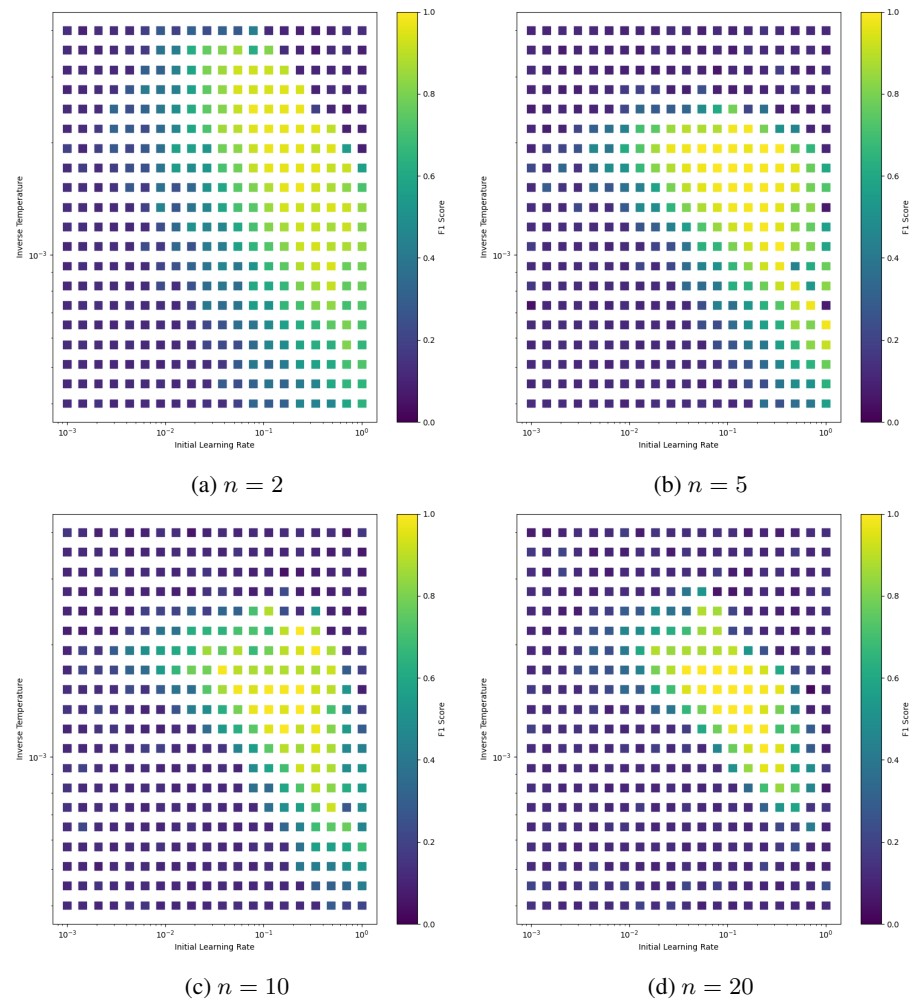

(a) $n = 2$

(b) $n = 5$

(c) $n = 10$

(d) $n = 20$

Figure 29: Hyperparameter search space for the original network, measuring the validation F1 score on the MNIST dataset. A larger F1 score corresponds to a better performing network.

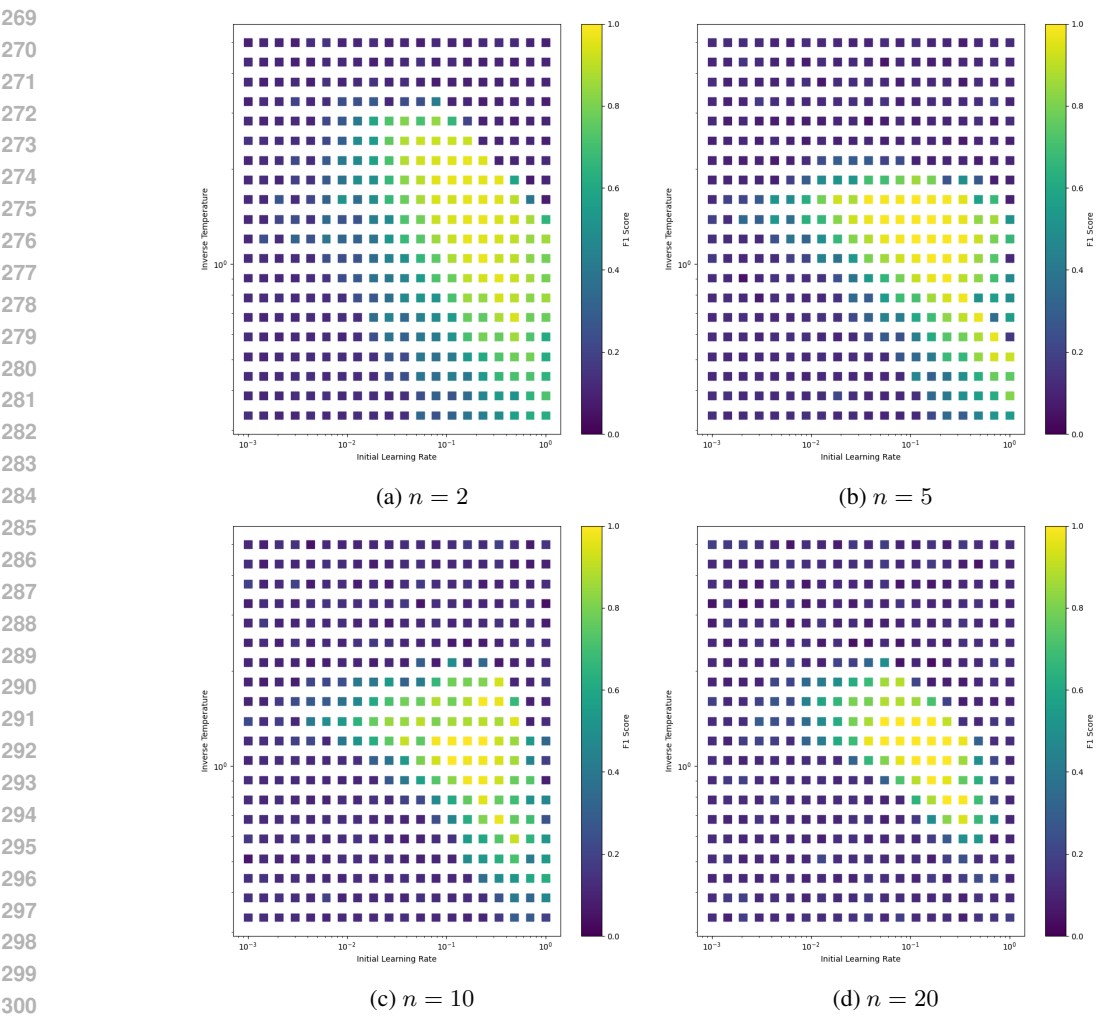

(a) $n = 2$

(b) $n = 5$

(c) $n = 10$

(d) $n = 20$

Figure 30: Hyperparameter search space for the modified network, measuring the validation F1 score on the MNIST dataset. A larger F1 score corresponds to a better performing network.

