# OpenReview forum: "Improved Robustness and Hyperparameter Selection in the Dense Associative Memory"
_ICLR.cc/2025/Conference — ICLR 2025 Conference Withdrawn Submission_

### Official Review · Reviewer_4Rvk · 2024-10-28

**Soundness:** 4
**Presentation:** 3
**Contribution:** 2
**Rating:** 5
**Confidence:** 5

**Summary:**

In the original work by [Krotov and Hopfield (2016)](https://arxiv.org/abs/1606.01164), they introduced a new energy function which allows for better separation of the stored attractors, such that they will not interfere with one another when the retrieval dynamic is conducted. This particular function is a polynomial function, that is based on a given power constant $n$. However, when the power $n$ is large (e.g., $n = 30$) and the number of neurons $N \rightarrow \infty$, numerical overflowing occurs. To combat this case, this work introduces a scaling factor $\alpha$ to the original update rule (specifically, $\alpha$ is multiplied to the dot product between the memories $\xi$ and the query) from [Krotov and Hopfield (2016)](https://arxiv.org/abs/1606.01164), which allows for stable computation when both $n$ and $N$ are large.

**Strengths:**

Based on the objective of the work, the approach is sound and is demonstrated to stablize the numerical issue for the update rule. The authors also show that by applying the scaling factor $\alpha$ to the dot product--- it does not change resultant of the update rule (e.g., the result of the sign function is not modified) accordingly to the homogeneity of the interaction (or polynomial) function.

The experimentation of the work illustrates that as the power $n$ increases, the modified model is still able to perform recovery unlike the original method from [Krotov and Hopfield (2016)](https://arxiv.org/abs/1606.01164), which is unable to perform recovery given $n > 25$.

**Weaknesses:**

Although the experiments do demonstrate better stability when the power $n$ is large, they are still quite insufficient.

Firstly, this paper lacks a comparison to Figure 1 of [Krotov and Hopfield (2016)](https://arxiv.org/abs/1606.01164), which depicts the test error for the number of epochs as the model is trained.

Secondly, there are no qualitative results of the memories just like those in Figure 2 of [Krotov and Hopfield (2016)](https://arxiv.org/abs/1606.01164). It is important to visually demonstrate these memories to check the effectiveness of memory storage via training.

Thirdly, the comparison, in the Appendices, only showcases the original network with dimension $N = 100$ and up to power value $n = 30$ starting from $n = 2$. The work should also illustrate its performance all the way to $n = 100$ or until numerical overflow occurs for a complete contrast.

**Questions:**

Does increasing $n$ always guaranteed that the performance will improve for both modified and original models?

Can we see a result similar to Figure 1 in [Krotov and Hopfield (2016)](https://arxiv.org/abs/1606.01164)? Additionally, a couple of visualizations of the trained memories $\xi$ as $n$ increases, would be very helpful!

Could you include the exact values for $\alpha$ and $\beta$ (and $T$) used in the experiments? I understand that it is trivial since $\alpha = 1/N$ and $\beta = \frac{1}{NT}$. But for transparency, it's best to include them anyhow.

---

### Official Review · Reviewer_sEkv · 2024-11-04

**Soundness:** 2
**Presentation:** 2
**Contribution:** 1
**Rating:** 3
**Confidence:** 5

**Summary:**

This paper shows that, by intelligently normalizing the similarity scores of a DenseAM, it is easier to find hyperparameters that successfully store patterns across increasingly "spiky" activation functions $F_{n}(x)$ while preventing numerical overflow for large $n$.

In more detail, the primary message of this paper is that:

1. "Similarity scores" (i.e., the dot product between any probe vector and the stored patterns) can be scaled by a constant before being passed to a "separation function" (e.g., the $n$-interaction vertex $F_{n}$) while maintaining the desirable properties of energy descent.
2. "Similarity scores" can additionally be scaled by an inverse temperature $\beta$ before being passed to the "separation function"

Both of these modifications can be condensed into multiplying the similarity scores of krotov&Hopfield by $\beta = \frac{1}{NT}$ [L338] to help stabilize DenseAM training by controlling the range of expected values and preventing numerical overflow.

**Strengths:**

(S1) **Easy to read and follow**. For the most part, the math in this paper is simply presented and easy to follow.

**Weaknesses:**

(W1) **Minimal novelty and contribution**. The paper studies only (A) the normalization of similarity scores (taken in this paper to be $\frac{1}{N} \zeta \dot \xi$ where $\zeta, \xi \in \mathbb{R}^{N}$, a simple numerical average) and (B) an inverse temperature $\beta$ that is placed inside the separation function $F_{n}$. Note that (B) is not novel, since this inverse temperature was placed inside the separation function is exactly what is done in the `LogSumExp` energy function from Ramsauer et al. (see Eq. (1)). That (A) the average dot product does not alter fixed point behavior is simply a property of a metric or "similarity score", as it is called in this paper. Additionally, normalization schemes mentioned in [L106-120] include a learnable scale and shift.

(W2) **Weak empirical results**. Memorization experiments are only trained to store "20 randomly generated bipolar vectors of dimension 100" $N=100$ [L354] (I do not see the results for the statement that "Larger dimensions and other dataset sizes were tested with similar results" [L355]). The claim of this paper is that it should now be easy to optimize hyperparameters for large $N$, but the results of this regime are not reported. Classification experiments only consider MNIST data ($N=784+10$). Fig. 1 could be condensed -- if the message is simply that their "similarity score" scaling choices allow memorization of patterns at high $n$, then you could define a scalar that measures the range of betas that could successfully memorize patterns across all tested learning rates, and plot $n$ (x-axis) vs. valid-range-of-$\beta$ (y-axis). This plot will go to 0 for high $n$ in the unscaled DenseAM.


**Summary**

The results in this paper are valid -- we are certainly allowed to scale similarity scores before passing them to separation functions. However, the contributions of the theory are not novel and the empirical results are not significant. With further characterization of the method (e.g., considering more separation functions e.g., the LogSumExp, and considering the spherical vectors of Ramsauer et al., and testing across more diverse datasets), I can see this paper as a good fit for a workshop. It is not ready for acceptance at ICLR.

**Questions:**

**Questions**

(Q1) [L081] "Shifting the scaling factors within the interaction function..." --> I assume the "shift" is due to the $\alpha$ and $\beta$ in Eq. (1) and (2). But these are multiplicative constants, which is a "scaling" operation. What do you mean by "shift"?

(Q2) [L087-088] Can you please elaborate on the following statement? This is not clear to me and it is not explained in the paper

> The feed-forward equivalent model implicitly implements our proposed changes by selecting values of the caling factor that negate terms arising from a Taylor expansion.

(Q3) [L031] states that all auto-associative memories suffer from capacity issues, but I thought the DenseAM remedies this?

**Comments**

(C1) [L260-270] This proof only only holds if constant $\alpha$ is positive, in which case the proof is banal since the only influence on the sign function is the difference of energies.

(C2) [L095-120] The paragraphs on normalization only serve to reveal the lack of novelty of this work. Indeed, LayerNorms include both a learnable scale and shift -- how is that distinct from the simple "average" dot-product similarity presented in this work?

(C3) [L064-075,140] The original DenseAM paper used $\sigma_{i}$ to denote the state vector and $\xi_{i}^{\mu}$ to denote the memory vectors. The notation of $\zeta_{i}^{\mu}$ as the memory vector and $\xi_{i}^{(t)}$ as the probe vector is possibly confusing to those familiar with the field.

---

### Official Review · Reviewer_fBzL · 2024-11-05

**Soundness:** 3
**Presentation:** 3
**Contribution:** 1
**Rating:** 3
**Confidence:** 3

**Summary:**

This paper proposes modifications to Dense Associative Memory (DAM) to address computational issues related to numerical instability and floating-point overflows when using large interaction vertices. By normalizing similarity scores and shifting the scaling factors inside the interaction function, the authors claim to preserve the network's dynamics while improving numerical stability and hyperparameter selection. They provide theoretical proofs for homogeneous interaction functions and present experimental results demonstrating improved performance and stability.

**Strengths:**

- The authors effectively address floating-point overflows and imprecisions, enabling the DAM to function correctly even with large network dimensions and interaction vertices, allowing for a larger capacity as an associative memory.
- This modifications result in a more consistent optimal hyperparameter region compared to the original DAM network, reducing the need for extensive hyperparameter searches when changing the interaction vertex.
- The authors provide mathematical proofs to demonstrate that the modifications preserve the network's essential properties for homogeneous interaction functions (such as polynomial functions) and even discuss how it applies to nonhomogenous interaction functions such as the exponential function.

**Weaknesses:**

- The technical contribution of this paper seems quite limited, simply moving the normalization factor inside of the dot product. Furthermore, I'm not sure if this is truly novel. This is mentioned on Page 2 where the feed-forward equivalent of the model already implicitly implemented the proposed changes. This kind of normalization might not have been explicitly mentioned in other papers, but I believe that Chaudhry et al. 2024 (Long Sequence Hopfield Memory) normalized the overlap between the network state and the stored patterns which achieve a similar result as the current paper. Krotov and Hopfield also mention various forms of normalization in their subsequent work.
- The theorems and proofs included in the main section of the paper are quite simple, and in reality should be moved to the appendix instead.
- The classification tasks chosen is exceedingly simple, limited to the MNIST dataset. It should be tested / demonstrated on more complicated datasets.

**Questions:**

- Test the modified DAM on larger and more complex datasets, such as CIFAR-10/100 or ImageNet, to showcase the practical benefits and scalability of the approach.
- The contribution of moving the normalization inside the interaction vertex seems limited, but I would be willing to change my opinion if it makes the DAM network performant for datasets / tasks that it was previously unable to solve altogether.
- How can this work be applied to concepts such as Hierarchical Associative Memory (Krotov 2021)?

---

### Official Review · Reviewer_FtfX · 2024-11-10

**Soundness:** 3
**Presentation:** 3
**Contribution:** 2
**Rating:** 3
**Confidence:** 4

**Summary:**

The work studies numerical stability and hyperparameter selection in Dense Associative Memory models. It is shown that the conventional formulation suffers from numerical issues, which result from applying steep interaction functions to arguments that are of order $N$ (size of the network). It is proposed to rescale these large numbers away from the model before applying the interaction function.

**Strengths:**

The paper is written clearly and the proposal is meaningful and makes sense. Indeed, Dense Associative Memories can be tricky to train, for reasons discussed in this work, and the problems studied in this work are important.

**Weaknesses:**

The main weakness is that the contribution is incremental. It is true that the pre-interactions in the model can be rescaled without breaking the desirable properties of Dense Associative Memories for homogenous interaction functions. At the same time, this submission looks more like a technical note rather than a full scale ICLR submission.

There is an error/typo in lines 268-269, last equality should lead to $e^{log(\alpha)+x} - e^{log(\alpha)+y}$. What is the conclusion from this subsection?

**Questions:**

What is the conclusion of subsection 4.1.1? The proposed rescaling would only work for homogeneous interaction functions, right?

---

### Author Response · Authors · 2024-11-20
**General Response to Reviewers**

We thank the reviews for their time and effort put into their reviews. We greatly appreciate the feedback provided and look to implement it into our manuscript. Due to the substantial amount of work required to update our paper with respect to the reviews we withdraw our paper from ICLR2025, but still appreciate the consideration given to us.

### On Novelty

Perhaps the most significant point made against our work is the apparent lack of novelty. It is true that other research has implemented some of our proposed modifications implicitly, but we believe it is for this reason that these other models work so well. The Dense Associative Memory (Krotov and Hopfield, 2016) has otherwise been superseded in literature by the Modern Continuous Hopfield Network and other physics inspired models of associative memory. Part of this is the ability for these other networks to operate over continuous domains, but as we show part of this is likely the instability of the DAM in comparison to other models. Our work shows explicitly why this instability is present in the DAM but not in these other models, something that the literature has not demonstrated. We believe that this type of fundamental research into the model architectures is novel and valuable to understanding and working with associative memories. In saying that we plan to justify this point much more in our manuscript to make our view obvious.

### On Continuous Data and Other Models

As pointed out by several reviewers, our results are only tested over bipolar datasets and not continuous domains. This was done to conform to the original DAM architecture which was not generalized to continuous data until the Modern Continuous Hopfield Network proposed by Ramsauer et al. (2020). We plan to investigate our modifications in the MCHN for completion, adding further to our results, but the main focus of our paper is most certainly on the original DAM. This is particularly because our work is on investigating the differences between the Hopfield network and the DAM, in which both networks most commonly operate over the bipolar domain.

Several reviewers expressed interest in seeing these modifications and results applied to other models, specifically those based on the DAM such as the MCHN, Hierarchical Associative Memories, and the Energy Transformer. We plan to investigate these model architectures further, expanding the scope of our work to include these subsequent models in our modifications.

We again than the reviews for their time.

---

### Note · Authors · 2024-11-20

I have read and agree with the venue's withdrawal policy on behalf of myself and my co-authors.